# Understanding the Epitranscriptome for Avant-Garde Brain Tumour Diagnostics

**DOI:** 10.3390/cancers15041232

**Published:** 2023-02-15

**Authors:** Ágota Tűzesi, Susannah Hallal, Laveniya Satgunaseelan, Michael E. Buckland, Kimberley L. Alexander

**Affiliations:** 1Department of Neuropathology, Royal Prince Alfred Hospital, Camperdown, NSW 2050, Australia; 2School of Medical Sciences, Faculty of Medicine and Health, The University of Sydney, Camperdown, NSW 2050, Australia; 3Department of Neurosurgery, Chris O’Brien Lifehouse, Camperdown, NSW 2050, Australia; 4Sydney Medical School, Faculty of Medicine and Health Sciences, The University of Sydney, Sydney, NSW 2050, Australia

**Keywords:** glioma, epitranscriptomics, m^6^A RNA methylation, RNA regulation, diagnostics

## Abstract

**Simple Summary:**

Glioblastoma is a complex and aggressive primary brain tumour that is rapidly fatal. Timely and accurate diagnosis is therefore crucial. Here, we explore the newly emerging field of epitranscriptomics to understand the modifications that occur on RNA molecules in the healthy and diseased brain, focusing on glioblastoma. RNA modifications are modulated by various regulators and are diverse, specific, reversible, and involved in many aspects of brain tumour biology. Epitranscriptomic biomarkers may therefore be ideal candidates for clinical diagnostic workflows. This review summarises the current understanding of epitranscriptomics and its clinical relevance in brain cancer diagnostics.

**Abstract:**

RNA modifications are diverse, dynamic, and reversible transcript alterations rapidly gaining attention due to their newly defined RNA regulatory roles in cellular pathways and pathogenic mechanisms. The exciting emerging field of ‘epitranscriptomics’ is predominantly centred on studying the most abundant mRNA modification, N6-methyladenine (m^6^A). The m^6^A mark, similar to many other RNA modifications, is strictly regulated by so-called ‘writer’, ‘reader’, and ‘eraser’ protein species. The abundance of genes coding for the expression of these regulator proteins and m^6^A levels shows great potential as diagnostic and predictive tools across several cancer fields. This review explores our current understanding of RNA modifications in glioma biology and the potential of epitranscriptomics to develop new diagnostic and predictive classification tools that can stratify these highly complex and heterogeneous brain tumours.

## 1. Introduction

Analogous to the epigenome, the epitranscriptome comprises a vast, dense, and complex web of chemical modifications on RNA molecules that facilitate the fine-tuning of gene expression regulation [1,2]. RNA modifications are reversible and highly controlled by protein regulators. These protein regulators include ‘writers’ that deposit chemical modifications on RNA molecules, ‘readers’ that recognise different RNA modifications, and ‘erasers’, which oppose the writers’ function by removing changes from RNA molecules [3,4]. There have been more than 100 RNA modifications identified to date, with the n6-methyladenine (m^6^A) modification being the most well studied, perhaps owing to its highest frequency of detection [5]. The m^6^A mRNA modifications are typically identified on 6–7 nucleotides per 3000-nucleotide-long mRNA and are presumed to have crucial regulatory roles [6,7]. While the exact downstream impacts of epitranscriptomic modifications on gene expression and biological function are not well understood, epitranscriptomic alterations are known to influence the nuclear transport of RNA molecules, the decay and translation of mRNA [8,9,10,11,12], and mRNA processing mechanisms such as splicing [13] and 3′ end processing, as well as miRNA processing [14,15,16,17].

Alterations in the genome, epigenome, and/or epitranscriptome are understood to underpin cancer molecular pathology, with implications for all aspects of cancer biology, including tumourigenesis and disease progression [18,19]. Glioblastoma (GBM), the most common and deadliest brain tumour in adults, is no exception [18,20,21]. While genomic and epigenomic alterations have been extensively investigated in GBM, the epitranscriptome remains largely unexplored.

Although RNA modifications in mammalian cells were first described almost 50 years ago [7], it is only until the recent emergence of new state-of-the-art technologies that large-scale explorations of epitranscriptomic modifications have been facilitated. It is now recognised that RNA modifications are diverse, dynamic, and of reversible nature, and this facilitates their role as fine tuners of RNA structure and function. There is also a growing appreciation of their regulation by various environmental factors. Therefore, a more thorough understanding of the epitranscriptome and its intricacies, particularly in cancer, will likely lead to innovative diagnostic tools and adjuncts for personalised therapeutics. In this review, we will discuss epitranscriptomics with a focus on m^6^A regulators and their roles, m^6^A modifications in the healthy and cancerous brain, and the current and future perspectives of epitranscriptomics in neurological disease and GBM diagnostics.

## 2. Epitranscriptomic Diversity and the Cellular Roles of RNA Modifications

There is great diversity among the RNA modification types, with more than 100 different chemical modifications reported in RNA species [22], which are precisely regulated by RNA-binding proteins. The most well-known RNA modifications, their epitranscriptomic regulators, and their roles are presented in Figure 1 and summarised below. Epitranscriptomic regulators and RNA modification types vary enormously, rendering biological pathways both complex and well regulated.

### 2.1. N6-Methyladenosine (m^6^A) RNA Modification

The methylation of adenosine (A) at the N^6^ position generates N^6^-methyladenosine (m^6^A), the most abundant and reversible epitranscriptomic modification in mammalian mRNA and noncoding RNAs (ncRNA). There are usually 1–3 m^6^A sites per mRNA molecule, which comprises approximately 0.2% of the overall transcript [7,23]. The m^6^A modification is preferentially found at RRACH positions (R = guanine (G) or adenosine (A); H = adenosine (A), cytosine (C), or uracil (U)) of consensus sequences [24], although this modification is not defined strictly by the presence of these sequences [25]. The distribution of m^6^A modifications occurs preferentially in different regions along the mRNA length, including around stop codons, long internal exons, and the 3′UTR end. m^6^A modifications at the 5′UTR end are less frequent; when present, m^6^A acts to promote mRNA translation at the 5′UTR [11,24,26]. Nevertheless, m^6^A distribution patterns along the mRNA sequence can change with environmental stressors, such as UV or heat shock [11].

#### 2.1.1. m^6^A Writers

The m^6^A methylation mark (in eukaryotes) is deposited by RNA methylation regulators (protein complexes with methyltransferase function) termed ‘writers’ (Figure 1B) during transcription in the nucleus. The m^6^A ‘writer’ complex comprises three main subunits—(1) methyltransferase-like 3 (METTL3), the catalytic subunit; (2) methyltransferase-like 14 (METTL14), the noncatalytic subunit that has a role in maintaining the integrity of this complex and ensuring RNA-binding [27]; and (3) Wilms’ tumour 1-associating protein (WTAP), which acts to recruit the methyltransferase complex to a target RNA by specifically binding to the Rm6ACH motif of mRNA molecules. The methyltransferase complex formed by METTL3, METTL14, and WTAP regulators preferentially targets internal adenosine residues and is termed WTAP-dependent methylation [15,28]; however, there are specific WTAP-independent methylation sites present around the RNA cap structure [28]. In addition to these subunits, there are proteins that interact with the m^6^A ‘writer’ complex and appear to be essential for methylation, such as KIAA1429. However, KIAA1429 knockdown experiments yielded a smaller reduction in methylation than WTAP-silencing experiments [28]. This knockdown investigation revealed the relationship between gene levels and methylation densities. Low-expression genes are more likely to be methylated than highly expressed genes; typically, ‘housekeeping’ genes lack methylation sites. WTAP-dependent mediated methylation is inversely correlated with mRNA stability [28], and through methyltransferase complex regulation, WTAP and METTL3 are implicated in the expression and alternative splicing of genes associated with transcription and RNA processing. Unexpected new evidence shows that METTL3 can also bind to non-m^6^A-modified RNA and enhance the translation of epigenetic factors [29].

In various coding and noncoding RNAs, i.e., pre-mRNAs and small nuclear RNA (snRNA), methyltransferase-like 16 (METTL16) was found to be a cytoplasmic RNA-binding ‘writer’ protein but also promoted translation in an m^6^A-independent manner [30,31,32,33]. In mouse embryonic stem cells (mESC), additional components of the ‘writer’ methyltransferase complex were identified, such as Zc3h13 and Hakai. These two components were found to interact with WTAP and Virilizer (the mouse homolog of KIAA1429, also named VIRMA in humans), suggesting the involvement of these components in the m^6^A methylation process [34]. In human cells, it was also shown that VIRMA interacts with WTAP/HAKAI/ZC3H13 to recruit METTL3 and METTL14 to the methyltransferase complex [35]. The interaction between HAKAI and WTAP is implicated in cell cycle regulation [36]. Two other possible interactor components of the ‘writer’ complex include the RNA-binding motif protein 15 (RBM15) and RBM15B, identified around the methylated DRACH motif. This suggests a role in recruiting additional writer proteins to specific sites in X-inactive specific transcript (XIST) RNA [37]. The m^6^A ‘writer’ complex is mainly comprised of METTL3, METTL14, WTAP, HAKAI, ZC3H13, VIRMA, RBM15, and RBM15B, while other diverse proteins (e.g., METTL16) with specific functions can have ‘writer’ role, as described in the literature to date.

#### 2.1.2. m^6^A Readers

The YTH protein family members contain the YTH domain that specifically recognises or ‘reads’ the m^6^A mark on mRNA molecules compared to unmethylated adenosine residues [38]. With affinity proteomic experiments, it was shown that three members of the YTH ‘reader’ protein family (YTHDF1, YTHDF2, and YTHDF3) selectively bind to m^6^A sites, suggesting a possible physical interaction between writers and readers [28]. YTHDF1 has binding sites around stop codons, which are regions enriched in m^6^A modification sites. The function of YTHDF1 in RNA translation was elucidated by knockdown in HeLa cells showing reduced translation efficiency and the number of ribosome-bound reads of the target genes. The YTHDF1 and YTHDF2 have targets that are regulated separately, of which 50% are in common, with YTHDF1 responsible for translating methylated mRNA while YTHDF2 mediates mRNA degradation [9,39]. Another ‘reader’ from the same family, YTHDF3, interacts with YTHDF1 to facilitate protein synthesis and with YTHDF2 to degrade methylated RNA. While these three members of the ‘readers’ (YTHDF1, YTHDF2, and YTHDF3) perform highly dynamic and reversible modulations of mRNA in the cytoplasm, another ‘reader’, YTHDC1, oversees methylated RNA transport from the nucleus to the cytoplasm [10,40]. YTHDC1 is also involved in mRNA splicing by promoting exon inclusion through the SRSF3 splicing factor recruitment and blocking the SRSF10 binding to mRNA [41]. The YTHDC2 is another ‘reader’ of the YTH protein family that contains RNA helicase domains and regulates its targets through a 3’→5’ RNA helicase activity [42]. YTHDC2 also facilitates the translation efficiency of m^6^A-marked targets [43]. In genetic disorders such as fragile X syndrome, the fragile X mental retardation protein (FMRP) has been found to bind to m^6^A sites and interact with YTHDF2 in the cerebral cortex of mice. The FMRP ensures RNA stability, while YTHDF2 promotes RNA decay. This suggests that FMRP has a ‘reader’ role and, through its interaction with YTHDF2, contributes to the fragile X syndrome phenotype, a common inherited intellectual disability known to be driven by the loss of functional FMRP [44].

In addition to the YTH protein family members, numerous other m^6^A readers have been characterised (Figure 1A). The eukaryotic initiation factor 3 (eIF3) is a specific ‘reader’ of m^6^A in the 5′UTR region, where it directly binds m^6^A to promote cap-independent translation [11]. Several heterogeneous nuclear ribonucleoproteins (hnRNPs) family members are also known as ‘readers’. One such m^6^A reader, HNRNPA2B1, has an essential role in primary microRNA (pri-miRNA) processing [17]. HNRNPA2B1 directly interacts with DGCR8, a miRNA processing complex (Microprocessor) member, to facilitate pri-miRNA processing into the precursor (pre)-miRNA form [17]. Interestingly, pri-miRNAs that have m^6^A marks deposited by METTL3 are more efficiently processed by the Microprocessor complex than those that are unmethylated [16]. Another RNA-binding protein, HNRNPC, is also a ‘reader’ involved in pre-mRNA processing, where HNRNPC binding is facilitated by m^6^A modifications to adjacent mRNAs and long noncoding RNA (lncRNA) structures [45]. Similarly, it was found that m^6^A modifications provide better accessibility for HNRNPG binding in a study that identified more than 10,000 m^6^A sites regulating this RNA-HNRNPG interaction, suggesting possible effects on expression changes and splicing [46]. RNA-binding insulin-like growth factor 2 (IGF2BP) protein family members (IGF2BP1/2/3) are also novel m^6^A methylation ‘readers’, which protect RNA from degradation by enabling translation and RNA stability. Some IGF2BPs target, stabilise, and translate oncogenic mRNAs, such as MYC [47], highlighting the role of m^6^A methylation and regulation in cancer cell biology.

#### 2.1.3. m^6^A ‘Erasers’

The m^6^A methylation mark can be removed by m^6^A demethylase enzymes, termed ‘erasers’ (Figure 1C). The obesity-associated protein (FTO) was one of the first m^6^A regulators to be identified for its essential role in demethylation; FTO knockdown was shown to increase m^6^A levels, whereas FTO overexpression decreased the relative m^6^A methylation in HeLa and 293FT cells [48]. Furthermore, FTO was found to mediate m^6^A demethylation on internal nucleotides along the RNA, as well as other demethylation activities in the cap region (m^6^A_m_) and in small nuclear RNA (snRNA; m^6^A and m6A_m_) and transfer RNA (tRNA; m^1^A). Interestingly, the FTO-mediated m^6^A demethylation significantly influences the levels of target transcripts that have internal m^6^A and a combination of both internal and cap region methylation but does not influence transcripts with only cap m^6^A_m_ modifications. FTO was shown to target nuclear and cytoplasmic m^6^A sites; however, it preferentially demethylates m^6^A_m_ modifications [49], thus reducing the stability of transcripts methylated in the 5′ cap region (m^6^A_m_) [50]. Another m^6^A demethylase, ALKBH5, was also found to remove m^6^A oxidatively and affect RNA metabolism and RNA export to the cytoplasm [8].

#### 2.1.4. Repelled Proteins

Some RNA-binding proteins are repelled by the m^6^A mark and instead preferentially bind nonmethylated adenosine, influencing RNA stability and degradation (Figure 1D). The two most well-known m^6^A-repelled proteins are the stress granule proteins, G3BP1 and G3BP2, that regulate mRNA stability and are involved in embryonic development [51]. ELAV-like RNA-binding protein 1 (ELAVL1), also known as human antigen R (HuR), is another RNA-binding protein repelled by m^6^A. It is a well-known RNA stabiliser protein. If an m^6^A mark is proximate to the ELAVL1 binding site on an mRNA molecule, it promotes the binding of ELAVL1, leading to RNA stabilisation. However, if the m^6^A mark is further from the ELAVL1-binding site, the spatial distance will not allow ELAVL1-binding and will result in RNA degradation [52,53].

### 2.2. Other RNA Modifications

#### 2.2.1. N6,2-O-Dimethyladenosine: m^6^A_m_ Modifications

N6, 2′-Odimethyladenosine (m^6^A_m_) methylation is abundant in human mRNA (92%) and occurs at the transcription start site just after the N7-methylguanosine (m7G) mRNA cap. M^6^A_m_ methylation is mediated by phosphorylated CTD interacting factor 1 (PCIF1), a cap-specific adenosine-N6-methyltransferase (CAPAM), and facilitates the translation of capped mRNAs [54,55] (Figure 1E). Where m^6^A modification is associated with RNA degradation, m^6^A_m_ modifications are important for RNA stabilisation. This methylation mark can also be catalysed by methyltransferase 4, N6-Adenosine (METTL4), a protein encoded by the *METTL4* gene. *METTL4* is well characterised in GBM as harbouring a high frequency of missense mutations [19,56]. The m^6^A_m_ mark can be removed by the m^6^A ‘eraser’ FTO and can be ‘read’ by decapping MRNA 2 (DCP2) [19].

#### 2.2.2. N1-Methyladenosine: m^1^A Modifications

M^1^A modifications play a critical role in tRNA stability and have been detected in mitochondrial transcripts, mRNA, and ribosomal RNA (rRNA). M^1^A modifications are added to tRNA by specific m^1^A writers, tRNA methyltransferase 61A (TRMT61A), and tRNA methyltransferase 6 noncatalytic subunits (TRMT6), and to mitochondrial tRNAs by tRNA methyltransferase 10C, mitochondrial RNase P subunit (TRMT10C), and tRNA methyltransferase 61B (TRMT61B) (Figure 1F). The erasers for this mark are alkB homolog 1, histone H2A dioxygenase (ALKBH1) and alkB homolog 3, and alpha-ketoglutarate dependent dioxygenase (ALKBH3) [57].

#### 2.2.3. 5-Methylcytosine (m^5^C) and 5-Hydroxymethylcytidine (hm^5^C) Modifications

DNA methylation is one of the most well-known epigenetic modifications that mainly occurs on cytosine residues (m^5^C) [58]. While this modification is less known in RNA, several recent studies describe the importance of m^5^C in diverse functions of mRNA, ncRNA, rRNA, and tRNA. m^5^C is enriched in the 5′UTR and 3′UTR regions, and it is regulated by writers (including NOP2/Sun RNA methyltransferase 1–7 (NSUN1–7) and tRNA aspartic acid methyltransferase (DNMT2)), readers (Aly/REF export factor (ALYREF) and Y-box binding protein 1 (YBX1)), and erasers (tet methylcytosine dioxygenase 1–3 (TET1–3) and ALKBH1) (Figure 1G) [59,60]. Similar to DNA, the m^5^C can be oxidized in RNA, resulting in 5-hydroxymethylcytidine (hm^5^C) [61]. M^5^C ‘erasers’ from the TET family have a dual role as hm^5^C ‘writers’.

#### 2.2.4. Adenosine-to-Inosine (A-to-I) Modifications

RNA sequences can undergo specific editing, resulting in the synthesis of protein isoforms that are different to what is encoded by the original genomic sequences [62]. Such editing events include base modifications through deamination reactions, i.e., uridine (U) substitution of cytidine (C) or adenosine (A) converted to inosine (I) [63]. The adenosine deaminase RNA specific (ADAR) family members perform adenosine-to-inosine RNA editing and are also known to have an epitranscriptomic ‘writer’ role; the I ‘writer’ proteins are ADAR, ADARB1, and ADARB2 (Figure 1H).

#### 2.2.5. Pseudouridine (Ψ) Modifications

Pseudouridine (Ψ), termed the fifth RNA nucleotide [19], is the 5-ribosyl isomer of uridine and was the first RNA modification to be discovered. Ψ is the most abundant RNA modification and can be found in diverse RNA types, including mRNA, rRNA, tRNA, small nucleolar RNA (snoRNA), and snRNA [64,65]. The Ψ modification is catalysed in eukaryotes by several pseudouridine synthases (Pus enzymes), and its most well-known writer is dyskerin pseudouridine synthase 1 (DKC1) (Figure 1I).

## 3. Epitranscriptomics in Neurobiology and Brain Cancer

Although the abundance and roles of some RNA modifications are still in dispute, the epitranscriptome is thought to have specific relevance to stem cell biology and neurobiology. Human genetic analysis and animal model studies have revealed that alterations in RNA regulatory programs have critical roles in the aetiology of neurogenerative disorders, intellectual disability, mental disorders, and brain cancers [59,66,67,68,69]. To date, m^6^A in fragile X syndrome is the most well-characterised interaction, and a growing number of studies have linked mutations in epitranscriptomic enzymes with intellectual disability [44]. m^6^A modifications are highly abundant in the brain and play essential roles in embryonic stem cell differentiation, neural development, and neurodevelopmental disorders [70]. The roles of m^6^A regulators in neural development, glioma stem cells (GSCs), glioma progression, and treatment are expanded on below.

### 3.1. m^6^A Regulators in Neural Development and Physiology

In mammals, the brain shows the highest abundance of m^6^A methylation in the body, where it is linked to several dynamic developmental and physiological processes such as neurogenesis, axonal growth, synaptic plasticity, circadian rhythm, cognitive function, and stress response [71]. Several studies show that m^6^A methylation is developmentally altered. Temporal-specific m^6^A-methylation features were identified in lncRNA in the developing cortex [72], and levels of m^6^A in mRNA were observed to remain low throughout embryogenesis but dramatically increase by adulthood [11,24,26]. A transcriptome-wide m^6^A methylation study revealed differences in the distribution of this epitranscriptomic modification, where m^6^A levels were twice as high in the mouse cerebellum compared to the cortex [73]. Moreover, the temporal regulation of the m^6^A methylation regulators METTL3, METTL14, WTAP, ALKBH5, and FTO is essential for the precise control of postnatal cerebellar development [74]. m^6^A methylation is also biased toward neuronal transcripts relative to glial cells, which are localised in axons and dendritic shafts, indicating their role in neuronal-activity-dependent gene regulation [71,75,76]. In addition to spatiotemporal and cell type preferences, m^6^A methylation is essential in governing direct reprogramming into neuronal cells [77] (Figure 2A).

During gliogenesis, when neural stem cell (NSC) populations are replaced by glial, astrocyte, and oligodendrocyte precursor cells (GPC, APC, and OPCs, respectively) [78], several m^6^A methylation-related changes take place. This was evidenced by *Mettl3* and *Mettl14* knockouts in embryonic mice brains, leading to ablation of RNA m^6^A levels and prolonged radial glia cell cycles, with the extension of cortical neurogenesis into postnatal stages [79]. Similarly, m^6^A signalling was also demonstrated to regulate neurogenesis in human forebrain organoids, where human ‘brain-disorder risk genes’ were observed to be enriched in m^6^A-tagged transcripts [79]. Dynamic RNA methylation also plays important regulatory roles in oligodendrocyte development and CNS myelination [80]. Conditional inactivation of *Mettl14* in mice disrupted oligodendroglial maturation with the selective depletion of oligodendrocytes where OPC numbers were within normal limits, with distinct effects on OPC and oligodendrocyte transcriptomes [80]. Oligodendrocyte specification and myelination are also influenced by epitranscriptomic regulators, such as m^6^A ‘reader’ Prrc2a (Proline rich coiled-coil 2 A) [81]. While key regulators of m^6^A methylation have been implicated in various neurological, neurodevelopmental, and neuropsychiatric disorders, m^6^A has specific roles in glioblastoma.

### 3.2. m^6^A Regulators in Glioma Stem-like Cells (GSCs) and Tumourigenicity

Glioblastomas are highly heterogeneous primary brain tumours consisting of diverse cellular populations, including glioma stem-like cells (GSCs). GSCs are a subset of distinct aberrant neural stem cells that possess glioma self-renewal potential and are thought to be responsible for driving GBM initiation, progression, treatment resistance, and recurrence. Interestingly, m^6^A mRNA methylation machinery was demonstrated to play a vital role in GSC self-renewal and tumourigenesis (Figure 2B) [82]. In knockdown experiments of m^6^A ‘writers’, METTL3 and METTL14 reduced m^6^A levels and dramatically enhanced GSC growth and self-renewal. Conversely, METLL3 overexpression or inhibition of the m^6^A ‘eraser’, FTO, increased m^6^A methylation levels and suppressed GSC self-renewal, proliferation, and tumour progression, resulting in prolonged survival of GSC-grafted mice [82]. To extrapolate, m^6^A methylation in GSCs may direct cell fate decisions, with higher m^6^A levels promoting GSC differentiation, whereas reduced m^6^A levels enhance GSC proliferation and tumourigenesis.

Indeed, GSCs and differentiated GBM cells have distinct patterns of m^6^A methylation [83]. However, a set of common transcripts from primary GSC spheroid cultures that were observed to lose RNA methylation during cell differentiation correlated with increased translation rates during the GSC transition to differentiated GBM cells. Moreover, miRNA-binding sites were observed to overlap with areas rich in RRACH motifs, i.e., m^6^A binding sites, which led to the identification of specific tumour-suppressive RRACH binding miRNAs that facilitate FTO-dependent, transcript-specific demethylation [83]. Taken together, this study showed that specific miRNAs may directly influence mRNA stability without engaging in expected miRNA mediated transcript downregulation, facilitating a fast and accurate adaptation to meet the cellular requirements of GSC differentiation [83].

The other m^6^A ‘eraser’, ALKBH5, has also been linked to GSC self-renewal and tumourigenesis [84] through its regulation of FOXM1 expression, a pivotal cell cycle regulation transcription factor that functions to maintain GSC properties [85] and is overexpressed and associated with poor survival in GBM [86]. ALKBH5 cooperates with a nuclear lncRNA, FOXM1-AS, to demethylate FOXM1 nascent transcripts and stimulate FOXM1 expression in GSCs, with the inhibition of ALKBH5 repressing GSC proliferation. In this process, ELAVL1, a nuclear RNA-binding protein that preferentially binds nonmethylated RNA, plays an important role in regulating FOXM1 expression by recruiting ALKBH5 to bind to the 3′UTR of unmethylated FOXM1 transcripts and promoting FOXM1 expression, contributing to GSC tumourigenicity [84].

Contrary to reports demonstrating positive associations between reduced m^6^A methylation and increased GSC tumourigenicity [82,84,87,88,89], several studies implicate an oncogenic role for METTL3 in cancer stem cells [90,91,92,93]. One such study purported that high levels of METTL3 and METTL3-dependent m^6^A modification are essential for GSC maintenance and dedifferentiation, with the preservation of a stem-like phenotype mediated by SOX2 mRNA stabilisation via the recruitment of ELAV1 to m^6^A-modified SOX2 [91]. The function of SOX2 as a transcription factor essential for embryonic and neural stem cell maintenance is well established. This study also implicates METTL3 in DNA repair efficiency, radiation resistance of GSCs, and survival outcomes in a GBM orthotopic mouse model.

Discrepancies between different studies might be explained by the diversity of cell types and m^6^A-modified RNA species analysed, as well as the intra- and intertumoural heterogeneity of GBM. Future experimental work must be context- and compartment-dependent to explain these inconsistencies and further our understanding of m^6^A RNA methylation and the interplay among regulator proteins in GSCs [91].

### 3.3. m^6^A Modifications and Regulators in Glioblastoma with Potential Therapeutic Implications for Improving Treatment Response

RNA modifications play increasingly important roles in the tumourigenesis and development of GBM; several ‘writers’, ‘erasers’, and even ‘readers’ of RNA modifications are proposed as potential diagnostic biomarkers and novel targets for treatment [19]. Yet, there are striking inconsistencies in m^6^A RNA methylation levels and m^6^A regulator expression reported in GBM [4,19]. In paediatric medulloblastoma, high METTL3 expression is associated with increased m^6^A levels and low survival, and a specific inhibitor of METTL3 slowed tumour progression in a medulloblastoma mouse model [94]. In line with the observed inverse correlation between GSC tumourigenicity and the general levels of m^6^A RNA methylation explored above [82], Li et al. reported that glioma tumours have lower global m^6^A RNA methylation levels relative to normal brain tissue, with decreased METTL3 and increased FTO levels in glioma tissues offered as a causative link [87]. Here, METTL3 overexpression in an established GBM cell line increased m^6^A levels and reduced cell proliferation and migration in vitro, perhaps related to a disruption in HSP90 chaperone activity and enhanced apoptosis [87]. Other studies corroborated the association between high METTL3 expression and reduced glioma proliferation; however, METTL3 silencing was shown to suppress vasculogenic mimicry, important for tumour angiogenesis [95,96]. Meanwhile, overexpression of another m^6^A writer, METTL4, was shown to promote glioma cell proliferation and migration [95,97]. Recently, FTO knockdown or inhibition was shown to suppress GBM proliferation via an m^6^A-dependent depletion of MYC levels [98]. Moreover, targeted FTO inhibition enhanced the efficiency of temozolomide (TMZ), the standard-of-care chemotherapy, in killing GBM cells [98]. Promisingly, FTO inhibition also shows antitumoural effects within tumour microenvironments, enhancing T cell-mediated cytotoxicity and reducing immune evasion by suppressing immune checkpoint genes [99]. While seemingly a promising novel therapeutic target, there are conflicting reports, including reduced FTO expression levels in more aggressive glioma tumour subtypes, with low FTO expression levels significantly associated with poor survival outcomes [100,101]. Similarly, specific inhibitors of another m^6^A ‘eraser’, ALKBH5, have been tested and show efficient inhibition of GBM cell proliferation [102] enhanced radiosensitivity and reduced invasiveness of GSCs [103], and improved anti-PD1 treatment efficiency in a preclinical glioma model [104]. While *ALKBH5* was reported as a significant negative prognostic factor for GBM patients [105,106], reported *ALKBH5* expression levels in GBM tissue are inconsistent [100,101,105,107].

Another approach to sensitise glioma cells to treatment was explored by modulating m^6^A ‘reader’ and member of the YTH protein family, YTHDF1. Relative to normal brain tissue, YTHDF1 has the highest expression in GBM, and is stabilised by musashi RNA-binding protein 1 (MSI1). GBM cell proliferation and migration are enhanced by the MSI1 stabilisation of YTHDF1, and high expression of these genes in GBM is associated with reduced survival [108]. Notably, YTHDF1 knockdown increased the sensitivity of glioma cells to the antiproliferative effects of TMZ, demonstrating its potential as a synergistic target for improved treatment response in GBM. Another YTH family member, YTHDF2, is also a potential target for glioma treatment, with studies linking YTHDF2 expression to glioma progression [97,109]. Increased in GSCs, YTHDF2 plays a vital role in stem maintenance, and it enhances tumour growth in mice through stabilising *MYC* (YTHDF2-MYC) and targeting IGFBP3. Interestingly, *YTHDF2* expressing GSC have an increased sensitivity to cell killing by linsitinib, an IGF1/IGF1R inhibitor. As NSC cell viability is unaffected by linsitinib treatment, this drug is proposed as a potentially specific anti-GBM treatment [110].

Epitranscriptomic regulators and RNA modifications are extensively involved in essential functions of the healthy brain and play various roles in brain disease. This is well shown in glioblastoma, where m^6^A can also have a role in treatment response. Unfortunately, there are several inconsistencies in the literature regarding the expression levels of the regulators and global m^6^A levels, and more in-depth studies are required to decipher the intricate web of RNA modifications in glioblastoma.

### 3.4. Other RNA Modifications Implicated in Glioblastoma Biology and Treatment

Beyond m^6^A methylation, other forms of RNA methylation also play important roles in GBM biology. For instance, the downregulation of m^6^A_m_ writer, PCIF1, promotes glioma cell proliferation and tumour growth in mice [111]. m^5^C methylated mature miRNAs are linked to poor GBM patient outcomes through a mechanism where this modification inhibits miRNA gene silencing [112]. hm^5^C methylation was found to have a significant role in GBM formation through a mechanism involving ‘eraser’ TET1, while TET2 and TET3 downregulation is linked to GBM tumourigenesis [113,114,115]. Transcript levels of I ‘writers’ ADAR, ADARB1, and ADARB2 are reduced in different grades and types of brain tumours [116]; ADAR2 was found to inhibit GBM cell growth, and an abnormally expressed splice variant supressed adenosine-to-inosine RNA editing [117,118]. Upregulation of the Ψ writer, DKC1, was also shown to promote glioma cell proliferation, migration, and invasion [119,120]. Likewise, a Ψ regulator, pseudouridine synthase 7 (PUS7), is highly expressed in glioma tissue, associated with poor survival, and PUS7 inhibitors could suppress tumour growth in mice [121].

## 4. Epitranscriptomics in Diagnosis

### 4.1. RNA Methylation Detection Methods

The field of epitranscriptomics is rapidly expanding and is anticipated to impact clinical diagnostics. The search for rapid, simple, and reliable methods to explore and detect new disease classification and therapeutic targets are essential steps for precision medicine. Currently, several RNA methylation detection methods exist, with diverse sensitivity, sample input requirements, and data analytical processes. Epitranscriptomics platforms may be antibody-based or employ mass spectrometry, polymerase chain reaction (PCR), next-generation sequencing (NGS), or nanopore direct RNA sequencing [122].

Probably, the simplest, most accessible, and affordable methods are antibody-based, i.e., m^6^A or m^5^C enzyme-linked immunoassay (ELISA). ELISA kits are useful for assessing global m^6^A/m^5^C methylation transcriptome-wide but are unable to distinguish m^6^A from m^6^A_m_ or detect a methylation mark at single-nucleotide or even at gene resolution. The detection of modifications at a single RNA species level can be measured by Dot-blot, another antibody-based method. This is also a relatively inexpensive method, with 100 ng to 1 μg required starting material, and can be used to detect RNA modifications in diverse RNA species [123,124].

Mass-spectrometry-based methods, i.e., liquid chromatography–coupled tandem mass spectrometry (LC–MS/MS), are more sensitive than antibody-based approaches. LC–MS/MS allows the absolute quantification of modified nucleosides, and, when coupled with nucleic acid isotope labelling (NAIL–MS), can assess the temporal dynamics of RNA modifications [125]. A drawback of LC–MS/MS is the high starting material needed (i.e., more than 1 μg of purified RNA), which may be unfeasible for some experimental setups [122]. Another method that uses isotope labelling in its workflow for RNA modification detection is 2D thin-layer chromatography (2D-TLC) [126]. This method is inexpensive and requires less starting material (50–200 ng) than LC–MS/MS; however, 2D-TLC can be adversely impacted by RNA digestion and labelling efficiencies [122].

PCR-based approaches are locus-specific RNA modification detection methods based on the premise that the queried modification impedes reverse transcription. These methods have high sensitivity and specificity; thus, they can be used for various RNA species. Some of the most well-known PCR-based approaches are the m^6^A-reverse transcription (RT)-quantitative PCR, single-base-elongation- and ligation-based PCR amplification method (SELECT), and primer-extension-based method [122,127].

NGS-based detection methods are diverse and frequently used for whole transcriptome profiling at single-nucleotide resolution. Some approaches require antibody coupling, such as m^6^A-seq or methylated RNA immunoprecipitation sequencing (MeRIP-seq) [26]. Other standard NGS-based detection methods are m^6^A individual-nucleotide resolution crosslinking and immunoprecipitation sequencing (miCLIP-seq), m6A-label-seq, m^6^A-sensitive RNA-endoribonuclease-facilitated sequencing (m6A-REF-seq), MazF RNase-assisted sequencing (MAZTER-seq), and deamination adjacent to RNA modification targets (DART-seq) [128,129,130,131]. More recent approaches include evolved TadA-assisted N6-methyladenosine sequencing (eTAM-seq), glyoxal- and nitrite-mediated deamination of unmethylated adenosines (GLORI), and m^6^A-selective allyl chemical labelling and sequencing (m^6^A-SAC-seq) [131,132,133,134,135]. m^6^A-SAC-seq requires very little starting material (2 ng of poly A^+^ RNA) but is a particularly laborious process.

One of the most promising epitranscriptomic approaches is Oxford Nanopore Technologies (ONT) sequencing, which can detect modifications in naïve RNA, such as m^6^A, inosine, or pseudouridine [136,137]. Compared to other methods, this method can detect various modifications in parallel from the same sample, directly from the RNA, without biases that can be introduced through reverse transcription required by NGS-based methods. The technology is based on the specific ionic current changes induced by single RNA molecules while passing through the nanopore. The current fluctuations differ between modified and nonmodified RNA molecules and can be decoded computationally [138]. Algorithms that can identify and analyse RNA modifications are fast evolving. Epitranscriptomic bioinformatic tools of note include—‘Explore the epitranscriptional landscape inferring from glitches of ONT signals’ (ELIGOS) that can predict known types of RNA modification sites, ‘ModPhred’ for user-friendly analysis of DNA and RNA modifications detected by ONT, and ‘MasterOfPores’, a processing pipeline for analysing direct RNA sequencing reads [139,140,141].

Beyond the m^6^A mark, other detection and data analytical methods facilitate the exploration of m^5^C, m^7^G, inosine, pseudouridine, and many other modifications on RNA. Although RNA modification detection methodology and bioinformatics tools are advancing rapidly, these techniques will require further adaptation before their implementation in diagnostic clinical service laboratories where there are real sample quantity and time constraints for analytical processes and data interpretation.

### 4.2. Epitranscriptomics in Current Diagnostic Approaches

The potential use of epitranscriptomic regulators as biomarkers has been explored across multiple cancer types; however, only a handful of studies have attempted to measure RNA modifications in body fluids directly. Of note, high global m^6^A levels are reported as potential diagnostic markers in the peripheral blood of small-cell lung (NSCLC) and breast and gastric cancer patients relative to healthy and/or benign cancer controls [142,143,144]. Here, m^6^A levels correlated to tumour-staging parameters were depleted following surgery, and accompanying upregulations in ‘writer’ and/or downregulations in ‘eraser’ genes were observed [142,143,144]. Likewise, m^6^A levels in rheumatoid arthritis (RA) patients’ and ischemic stroke patients’ blood were elevated relative to healthy controls, along with decreased ‘eraser’ ALKBH5 and FTO levels and increased YTHDF2 ‘reader’ levels in RA samples [145,146]. In contrast, m^6^A levels in the peripheral blood of individuals with Type 2 diabetes mellitus (T2DM) are lower than in controls, with concomitantly high m^6^A ‘eraser’ FTO gene expression [147]. Similar trends are observed in ageing, COVID-19 infection, and smoking [148,149,150]. A summary of studies investigating global m^6^A methylation status and gene expression of m^6^A regulators as diagnostic tools across a range of pathologies are presented in Table 1.

In recent epitranscriptomics studies exploring new diagnostic classification markers, investigations often stem from in silico analyses of public sequencing datasets (i.e., TCGA, CCGA) to identify the predictive and prognostic significance of the diverse regulators of RNA modifications. For instance, in lung adenocarcinoma, high HNRNPC and IGF2BP3 levels correlate with reduced overall survival [156,157], and a six-gene risk signature (KIAA1429, ALKBH5, METTL3, HNRNPC, YTHDC2, and YTHDF1) strongly correlated with various clinical and pathological features [158]. A further study also identified METTL3, YTHDF1, and YTHDF2 as prognostic lung adenocarcinoma biomarkers [159]. In lung squamous cell carcinoma, reduced KIAA1429, ALKBH5, METTL3, and HNRNPC expressions were predictive of better responses to chemotherapy and immunotherapy. Identified m^6^A regulators with significant prognostic value in a diverse range of cancer types are presented in Table 2. There is no clear consensus for a pan-cancer epitranscriptomic regulation profile but rather a range of epitranscriptomic players that are differentially modulated across conditions, suggesting that the expression of epitranscriptomic patterns is cancer subtype-specific.

### 4.3. Epitranscriptomics in Glioma Diagnostics

Like other cancer types, the value of epitranscriptomics as a tool for brain cancer diagnostics is still in its early days. The WHO classification system for primary brain tumours has integrated molecular diagnostics with traditional histopathology [197]. While the identification of molecular alterations, such as isocitrate dehydrogenase *IDH1/2* mutations, 1p/19q codeletion, and *MGMT* promoter methylation, has improved the accuracy of diagnostics, prognostication, and prediction of treatment response for glioma patients, there has been no tangible improvement in clinical management and patient outcomes. Epitranscriptomics of known glioma sub-entities may yield further advances in tumour stratification and help guide treatment selection for precision medicine, as was shown in preliminary studies of colon cancer [152]

Several studies explored m^6^A RNA methylation regulators in public primary brain cancer sequencing datasets, and these key findings are summarised in Table 3. Many of these studies are not based on integrated brain tumour diagnostic definitions (WHO 2021 classification), and tumours are often referred to in broad terms (GBM, glioma, astrocytoma, or low-grade glioma, LGG). In this review, we use the diagnostic terms used by the original studies; however, caution must be taken when interpreting RNA modification changes relevant to specific glioma subtypes.

Key findings in glioma patients in relation to m^6^A ‘readers’ include YTHDF2 overexpression, induced through the EGFR/SRC/ERK pathway to promote tumourigenesis, invasiveness, and cell proliferation independently associated with poor prognosis [198]. Similarly, expression of the YTHDF1 paralog is negatively associated with survival and promotes glioma cell proliferation and growth in vitro [100]. A systematic meta-analysis of m^6^A ‘reader’ eIF3 was performed in independent glioma cohorts as several eIF3 subunits are localised to chromosomes 1p and 19q [199,200], regions co-deleted in IDH-mutant oligodendroglioma [201]. All 13 eIF3 subunits were significantly differentially expressed between the glioma subgroups in both TCGA and CGGA datasets [202]. Among other findings, eIF3i expression was shown to be an independent prognostic factor in IDH-mutant LGG and could also predict the 1p/19q codeletion status [202]. m^6^A regulators *IGF2BP2*, *IGF2BP3*, *HNRNPC*, and *YTHDF2* and m^5^C-related long noncoding RNAs *CIRBP-AS1*, *GDNF-AS1*, *LINC00265*, and *ZBTB20*-*AS4* have prognostic significance in LGG [203,204].

Expression of the m^6^A ‘writer’ METTL3 was positively associated with a higher malignant grade and poorer prognosis of IDH-wildtype but not IDH-mutant gliomas [205]. Another study identified significant fluctuations in m^6^A regulator expression when comparing glioma cohorts comprising different IDH mutational and 1p/19q codeletion states. While these cohorts represent different glioma sub-entities with distinct molecular pathophysiology, the average survival times are vastly different, particularly between oligodendroglioma (IDH-mutant, 1p/19q co-deleted) and GBM (IDH-wildtype, 1p19q non-co-deleted), and thus divergent epitranscriptomic regulator expression patterns may relate to the very different biology of these distinct pathological entities [101]. In this study, two glioma subgroups were identified based on the expression of m^6^A regulators, and a risk signature (*ALKBH5*, *IGF2BP3*, *KIAA1429*, and *YTHDF2)* was significantly associated with prognosis, the immune microenvironment and treatment efficacy [101]. A similar risk signature (*ALKBH5*, *IGF2BP2*, *IGF2BP3*, *HNRNPA2B1*, *YTHDF1*, *YTHDF2*, *RBM15*, and *WTAP*) was shown to be prognostic for glioma patients, and the expression of *IGF2BP2* and *IGF2BP3* was linked to tumour occurrence, development, and progression [206]. A recent study also reported independent prognostic significance of *ALKBH5,* which is upregulated in glioma and associated with immune signalling, and inflammatory and metabolic pathways genes [207]. Another study calculated significant prognostic values for four hub genes *(EMP3*, *PDPN*, *TAGLN2,* and *TIMP1*) related to m^6^A regulators, all with high expression in high-grade glioma, and further correlation to IDH status and transcriptome subtype [208]. Links between clinical outcomes and m^6^A regulators function in the tumour microenvironment, tumour cell stemness, and infiltration have also been established [209]. Likewise, epitranscriptomic regulation of lncRNAs are also associated with glioma patient prognosis [210]. Further studies highlighting the close link between m^6^A RNA methylation regulators and clinicopathological features and treatment sensitivity of gliomas are outlined in Table 3.

**Table 3 cancers-15-01232-t003:** m^6^A regulators identified as biomarkers in glioma.

Increased m^6^A Regulator(s)	Decreased m^6^A Regulator(s)	Data Used	Observations/Role	Ref.
YTHDF2		TCGA, REMBRANDT French, Kawaguchi, Paugh	YTHDF2 is linked to glioma malignancy and invasiveness.	[198]
YTHDF2, YTHDF1, METTL3, RBM15, HNRNPC	ALKBH5, WTAP, YTHDC2, ZC3H13 METTL14, FTO	TCGA	YTHDF1 overexpression correlates with the advanced stage of disease. YTHDF1 contributes to glioma progression.	[100]
eIF3e		Oncomine, TCGA	eIF3 subunits show varied expression in distinct regions of GBM tumours. eIF3e proteins expression correlates with glioma grade, highest expression in GBM, and increases in recurrences. eIF3e upregulation in recurrences may have a role in treatment resistance.	[211]
eIF3b, eIF3i, eIF3k and eIF3m (poor OS)	eIF3a and eIF3l (better OS)	CGGA, TCGA	Expression of eIF3d, eIF3e, eIF3f, eIF3h, and eIF3l correlates with the IDH-mutant status of gliomas. eIF3i and eIF3k expressions increase with tumour grade and are associated with poor OS. eIF3i is an independent prognostic factor in IDH-mutant LGG and can predict the 1p/19q codeletion status in IDH-mutant LGG. High eIF3i expression correlates with cell proliferation, mRNA processing, translation, T-cell receptor signalling, NF-kB signalling, and many others.	[202]
METTL3		CGGA, TCGA	METTL3 promotes the malignant progression of gliomas in vitro and in vivo. METTL3 correlates with poor OS in IDH-wildtype but not in IDH-mutant gliomas.	[205]
eIF3A, FMR1, FTO, METTL14, METTL16, METTL3, RBMX, YTHDC, YTHDF3 and ZC3H13 (IDH-mutant vs. IDH-wildtype)	ALKBH5, IGF2BP2, IGF2BP3, RBM15, WTAP and YTHDF1 (IDH-mutant vs. IDH-wildtype)	CGGA, TCGA, REMBRANDT	Expression of m^6^A regulators is associated with Prognosis, grade, IDH, and 1p/19q status. Lower m^6^A regulators expression (except FTO) is associated with longer OS. A prognostic risk signature—ALKBH5, IGF2BP3, KIAA1429, and YTHDF2.	[101]
ALBKH5, RBM15, YTHDF and WTAP (increased tumour grade)	FTO	CGGA, TCGA	RBM15, METTL3, METTL14, ALKBH5, FTO, YTHDC1, and YTHDF2 are significantly differentially expressed between IDH-mutant and IDH-wildtype LGG. METTL3, FTO, and YTHDC1 are significantly differentially expressed between IDH-mutant and IDH-wildtype GBM. The risk signature comprises RBM15, WTAP, ALBKH5, FTO, YTHDC1, YTHDF1, and YTHDF2, all of which are independent prognostic markers and predictive of clinicopathological features and treatment sensitivity.	[105]
RBM15, RBM15B, METTL3, METTL14, WTAP, HNRNPA2B1, HNRNPC, YTHDF1, YTHDF2, YTHDF3, and YTHDC2 (gliomas vs. control) ALKBH5, RBM15, WTAP and YTHDF2 (in GBM vs. LGG)	FTO and ZC3H13 (gliomas vs. control) FTO, KIAA1429, METTL3, ZC3H13, HNRNPC, and YTHDC2 (in GBM vs. LGG)	CGGA	Four m^6^A-related lncRNAs that have prognostic values: LINC00900 and MIR155HG, increased in higher-grade tumours, while MIR9-3HG and LINC00515 have lower expression in HGG vs. LGG.	[107]
IGF2BP3	YTHDC2	TCGA, CGGA cBioportal	IGF2BP3 expression increases with tumour grade and correlates with shorter OS. YTHDC2 and IGF2BP3 are negative and positive prognostic factors for OS.	[212]
HNRNPC, WTAP, YTHDF2 and, YTHDF1		TCGA	Defined prognostic risk signature: HNRNPC, ZC3H13, and YTHDF2. HNRNPC plays an important role in malignancy and contributes to the development of gliomas. High expression of HNRNPC correlates with a favourable prognosis.	[213]
LINC00265	C6orf3, GDNF-AS1, LINC00925, LINC00237	TCGA, CGGA	Twenty-four prognostic m6A-related lncRNAs were identified as prognostic lncRNAs. m6A-related lncRNA prognostic signature (m6A-LPS).	[107]
METTL14, IGF2BP2, IGF2BP3, HNRNPA2B1, YTHDF1, YTHDF3,HNRNPC, RBMX, WTAP, YTHDF2, and IGF2BP1		TCGA, CGGA	Defined prognostic risk signature: ALKBH5, IGF2BP2, IGF2BP3, HNRNPA2B1, YTHDF1, YTHDF2, RBM15, and WTAP.	[206]

Again, there are several conflicting reports of m^6^A regulator expression levels among the different brain tumour entities and non-tumour brain tissue, including ALKBH5, WTAP, METTL14, and YTHDC2 [100,101,105,107,213]. The contradictory findings of m^6^A RNA methylation regulating gene expression levels may help to explain the opposing reports of high and low global m^6^A in brain tumour tissue relative to ‘normal’ brains [4,87]. There is some consensus, however, for the relative expressions of several m^6^A regulator genes, which are summarised in Table 4. Of these, one of the most discussed is m^6^A ‘erasers’ FTO, which is frequently observed to be downregulated in GBM relative to other tumours and control tissue [100,101]. If taken to be true, this suggests that GBM tumours have increased global m^6^A levels.

## 5. Conclusions and Future Perspectives

Epitranscriptome profiling studies, particularly in cancer research, are becoming more common. Studies investigating global m^6^A, m^5^C, and other RNA methylation profiles have helped to identify epitranscriptomic regulators and their involvement in cancer biology. While the direct assessments of RNA modifications in diagnostic laboratory settings are technologically challenging, there is a wealth of evidence supporting the significant relationships between epitranscriptomic regulator gene expressions and clinicopathological parameters in cancer, including their use as diagnostic, prognostic and/or predictive biomarkers. With distinct roles defined in stem cell biology and neural development along with high levels detected in brain tissue, RNA modifications are, theoretically, highly relevant to multiple aspects of gliomagenesis and biology. Indeed, multiple studies have reported epitranscriptome regulator signatures that are highly predictive of glioma pathology and patient outcomes.

Further technological advances are needed to adapt epitranscriptomic approaches for clinical diagnostics and will likely uncover entirely novel targets for further stratifying tumour molecular phenotypes, directing treatment(s) and guiding patient management. In the interim, the detection of regulator genes and global m^6^A methylation levels in highly annotated cohorts of discrete tumour types as well as biofluids are warranted.

## Figures and Tables

**Figure 1 cancers-15-01232-f001:**
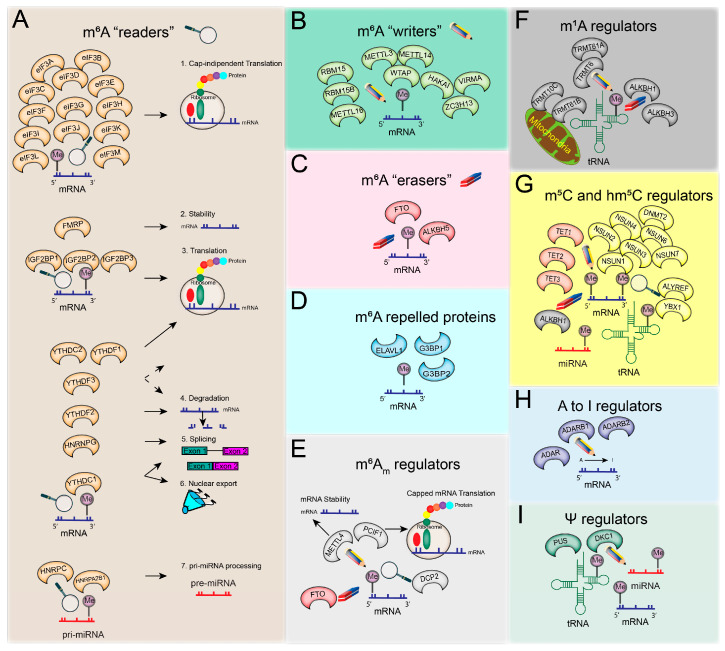
RNA modifications are mediated by known epitranscriptomic regulators: readers (magnifying glass), writers (pencil), and erasers (rubber). The panels depict (**A**) m^6^A reader members with their roles in different aspects of RNA biology; (**B**) m^6^A writers; (**C**) m^6^A erasers; (**D**) proteins repelled by m^6^A methylation; (**E**) m^6^A_m_ regulators and their known roles; (**F**) m^1^A regulators and their main role during tRNA and mitochondrial tRNA life; (**G**) m^5^C and hm^5^C regulators; (**H**) known adenosine-to-inosine (**A**–**I**) regulators; (**I**) Ψ regulators.

**Figure 2 cancers-15-01232-f002:**
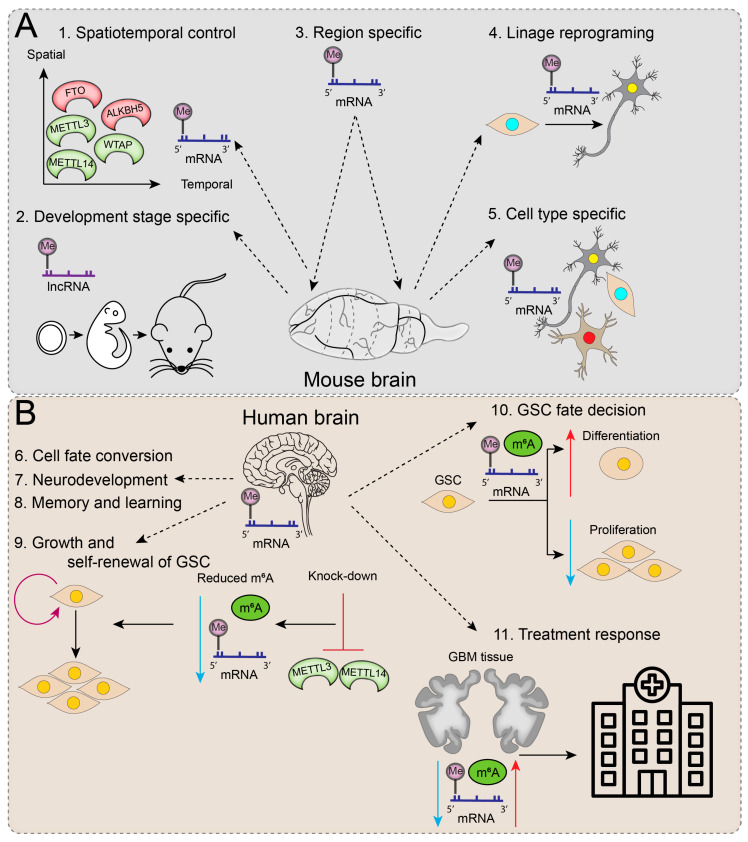
Functions of epitranscriptomic regulators and RNA modifications in the brain. (**A**) In an animal model (mouse), RNA methylation is predominantly related to spatiotemporal control and is specific to developmental stage, region, lineage, and cell type. (**B**) The main functions attributed to RNA methylations in the human brain are associated with cell fate conversion, neurodevelopment, memory, and learning. The knockdown of m^6^A ‘writers’ (METTL3 and METTL4) causes a decrease in m^6^A methylation, which is thought to play a role in GSC growth and cell renewal. The m^6^A methylation levels can influence the GSC fate decision towards differentiation (increase; red upward arrow) or proliferation (decrease; blue downward arrow). RNA modifications detected in GBM tissue reflect treatment-related responses, which could have great importance for clinical monitoring of GBM.

**Table 1 cancers-15-01232-t001:** m^6^A regulators expression direction and global m^6^A methylation status with biomarker value detected from blood. Increased and decreased m^6^A levels are depicted by up and down arrows, respectively.

Disease/Condition	Upregulated	Downregulated	m^6^A Levels	Ref.
Non-small-cell lung carcinoma	METTL3, METTL14, RBM15	ALKBH5, FTO	↑	[142]
Breast cancer	METTL14	FTO	↑	[143]
Gastric cancer		ALKBH5, FTO	↑	[144]
Acute myeloid leukaemia	WTAP			[151]
Colorectal cancer	IGF2BP2		↑	[152]
Rheumatoid arthritis	ALKBH5, FTO, YTHDF2		↑	[145]
Diabetes	FTO		↓	[147]
Spinal cord injury		FTO, METTL14, RBMX, YTHDF2, YTHDC2, HNRNPA2B1		[153]
Systemic lupus erythematosus		METTL3, METTL14, WTAP, FTO, ALKBH5, YTHDF2		[154]
COVID-19	METTL3, FTO		↓	[149]
Aging			↓	[148]
Smoking and air pollution			↓	[150]
Myocardial infarction (in rats)			↑	[155]
Ischemic stroke			↑	[146]

**Table 2 cancers-15-01232-t002:** Expression of epitranscriptomic regulators and their observed roles in diverse conditions.

Disease/Condition	RNA Modification or Epitranscriptomic Regulator	Sample Type	Observations/Role	Ref.
Smoking and air pollution	Global m^6^A levels	Peripheral blood	Decreased global m^6^A levels were found in smokers compared to non-smokers.	[150]
Aging	Global m^6^A levels	PBMCs	Decrease in overall m^6^A with aging. m^6^A-modified transcripts have higher expression than the nonmodified ones. DROSHA and AGO2 have high methylation levels in younger subjects.	[148]
COVID-19	m^6^A levels, METTL3, FTO	PBMCs	Increased METTL3 and FTO in COVID-19 patients. m^6^A modification has an essential role in the clinical status of COVID-19 patients	[149]
Spinal cord injury (SCI)	FTO, METTL14, RBMX, YTHDF2, YTHDC2, HNRNPA2B1	PBMCs, GEO data	METTL14, FTO, RBMX, YTHDF2, HNRNPA2B1, and YTHDC2 downregulated in SCI. AKT2/3 and PIK3R1 are potential m^6^A-related therapeutic targets.	[153]
Myocardial infarction (MI)	5mdC, 5mrC, m^6^A levels	Heart tissue and blood (rats)	Increased levels of 5mdC, 5mrC, and m^6^A in heart tissue eight weeks after surgery.	[155]
Rheumatoid arthritis	ALKBH5, FTO, YTHDF2, m^6^A levels	Peripheral blood	Decreased ALKBH5, FTO, and YTHDF2 as risk factors. Global m^6^A is increased and negatively correlates with decreased FTO gene expression.	[145]
Type 2 diabetes (T2DM)	FTO	Peripheral blood	m^6^A levels are lower in T2DM patients, probably caused by the higher FTO expression. Higher FTO level is associated with T2DM risk.	[147]
Systemic lupus erythematosus (SLE)	METTL3, WTAP, FTO, ALKBH5, YTHDF2	Peripheral blood	Decreased METTL3, WTAP, FTO, ALKBH5, and YTHDF2 gene expression in the blood of SLE patients. ALKBH5 as a risk factor and involved in SLE pathogenesis.	[160]
Systemic lupus erythematosus (SLE)	METTL14, ALKBH5, YTHDF2	PBMCs	Decreased expression of METTL14, ALKBH5, and YTHDF2 in SLE.	[154]
Endometriosis	FTO, HNRNPC, HNRNPA2B1	GSE6364 data	FTO, HNRNPC, and HNRNPA2B1 have biomarker potential.	[161]
Lung adenocarcinoma	HNRNPC, METTL3, YTHDC2, KIAA1429, ALKBH5, YTHDF1	Tissue	HNRNPC, METTL3, YTHDC2, KIAA1429, ALKBH5, and YTHDF1 linked to clinical features, pathological stages, gender, and survival.	[158]
Lung adenocarcinoma	HNRNPC	Tissue	HNRNPC high expression correlates with gender, age, ethnicity, lymph node metastasis, smoking history, TNM staging, and poor prognosis.	[156]
Lung adenocarcinoma	IGF2BP3	Tissue	IGF2BP3 correlates with poor prognosis, tumour length, differentiation, T stage, and gender. IGF2BP3 is an independent prognosis factor and potential oncogene.	[157]
Lung adenocarcinoma	KIAA1429, RBM15, METTL3, HNRNPC, HNRNPA2B1, YTHDF1, YTHDF2	Tissue	METTL3, YTHDF1, and YTHDF2 are prognostic biomarkers and suggest better OS and RFS.	[159]
Lung adenocarcinoma	METL3, VIRMA, RBM15, YTHDF1, YTHDF2, LRPPRC, HNRNPA2B, IGFBP3, RBMX, FTO, ALKBH5, WTAP, METTL16, METTL14, ZC3H13	Tissue	Risk factors that predict worse prognosis.	[162]
Non-small-cell lung carcinoma (NSCLC)	m^6^A levels, METTL3, METTL14, RBM15, ALKBH5, FTO	Peripheral blood	Leukocyte m^6^A levels potential biomarker for NSCLC screening, diagnosis, and monitoring.	[142]
Non-small cell lung cancer (NSCLC)	YTHDC2, METTL3, RBM15, HNRNPC, YTHDF2, YTHDF1, ZC3H13	Tissue	Gene signatures classify prognostic groups.	[163]
Non-small cell lung cancer (NSCLC)	HNRNPC	Tissue	HNRNPC predicts poor prognosis and correlates with lymph node metastasis and tumour invasion.	[164]
Lung squamous cell carcinoma	ALKBH5, METTL3, HNRNPC, KIAA1429	Tissue	T follicular helper cells have a prognostic signature role in predicting the survival and treatment response.	[165]
Breast cancer	METTL14, FTO, m^6^A levels	Peripheral blood	m^6^A levels are elevated in advanced tumour stages.	[143]
Hepatocellular carcinoma (HCC)	ALKBH5	Tissue and cells	Loss of ALKBH5 is an independent prognostic factor. ALKBH5 inhibits HCC proliferation in vitro and in vivo.	[166]
Hepatocellular carcinoma (HCC)	YTHDF1	Tissue	YTHDF1 is upregulated in HCC and correlates with stage. Lower YTHDF1 expression results in better OS. YTHDF1 is involved in HCC cell cycle progression and metabolism regulation.	[167]
Hepatocellular carcinoma (HC)	YTHDF1, YTHDF2, METTL3, KIAA1429	Tissue	m^6^A regulators differentially expressed in HC. Independent prognostic risk signature: YTHDF1, YTHDF2, METTL3, and KIAA1429.	[168]
Hepatocellular carcinoma (HCC)	METTL14	Tissue	METTL14 expression correlates with the expression and regulates m^6^A methylation of hub genes, CSAD, GOT2, and SOCS2.	[169]
Hepatocellular carcinoma (HCC)	m^5^C-related lncRNAs	Tissue	Prognosis value established for 8 m^5^C-related lncRNAs.	[170]
Gastric cancer (GC) and benign gastric disease (BGD)	m^6^A levels, ALKBH5, FTO	Peripheral blood	m^6^A levels are elevated in advanced tumour stages. m^6^A decreases after surgery. FTO in stage IV disease < stage I.	[144]
Gastric cancer	m^6^A levels, YTHDF1	Tissue and cell	Constructed a diagnostic m^6^A score that can distinguish cancer from normal tissue. YTHDF1 expression correlates with high-risk subtype patients, and it is a possible oncogene.	[171]
Gastric cancer (GC)	m^6^A levels	Tissue	m^6^A score is an independent prognostic biomarker. Lower m^6^A scores have EBV and MSI patients that are sensitive to checkpoint immunotherapy. Negative correlation between m^6^A score and mutation. EMT has the lowest m^6^A score.	[172]
Gastric cancer (GC)	m^6^A levels, METTL3	Tissue	High METTL3 expression increases m^6^A levels and is associated with GC proliferation, liver metastasis, and poor prognosis.	[173]
Gastric cancer (GC)	FTO, ALKBH1	Tissue	High expression of FTO and ALKBH1 transcripts associated with low survival. Low ALKBH1 protein expression correlates with larger tumour size and advanced TNM stages. Low FTO protein expression correlates with shorter OS.	[174]
Gastric cancer (GC)	RBM15, WTAP, METTL3, YTHDF2, YTHDF1, YTHDC1, YTHDC2, KIAA1429, ZC3H13, HNRNPC	Tissue	Hub genes associated with m^6^A regulators have prognostic values: AARD, ASPN, SLAMF9, MIR3117, and DUSP1. ASPN is also upregulated in GC cells.	[175]
Pancreatic cancer	KIAA1429, HNRNPC, METTL3, YTHDF1, IGF2BP2, IGF2BP3	Tissue, cell line	m^6^A-regulator risk signature.	[176]
Pancreatic adenocarcinoma	RBM15	Tissue	Various significant prognostic parameters.	[177]
Colonic adenocarcinoma	YTHDF1, METTL3, KIAA1429, YTHDF3, YTHDC2, METTL14, ALKBH5	Tissue	YTHDF1, YTHDF3, and YTHDC2 are promising biomarkers for detection, progression, and prognosis.	[178]
Colon cancer	ALKBH5	Tissue, cells	ALKBH5 has a tumour suppressor role in CC. Overexpression of ALKBH5 can inhibit CC invasion and metastasis and has prognostic significance.	[179]
Colorectal cancer (CRC)	m^6^A levels, IGF2BP2	PBMCs, GEO data	m^6^A in the blood is a prospective biomarker for CRC and a possible therapeutic target. IGF2BP2 has high expression in CRC blood. Monocytes have the most m^6^A modification.	[152]
Colorectal adenocarcinoma	METTL3, YTHDF1, IGF2BP1, IGF2BP3, EIF3B, HNRNPA2B, YTHDF1, IGF2BP1, IGF2BP3	Tissue	Potential biomarkers YTHDF1, IGF2BP1, IGF2BP3, and EIF3B.	
Acute myeloid leukaemia	WTAP	Peripheral blood, bone marrow cells	Patients were classified into two risk groups based on WTAP expression. High WTAP more common in older patients.	[151]
Acute myeloid leukaemia	ZC3H13, RBM15, LRPPRC, METTL14, YTHDC2	Tissue	METTL14, YTHDC2, ZC3H13, and RBM15 expression correlates with OS.	[180]
Neuroblastoma	METT14, WTAP, HNRNPC, YTHDF1, IGF2BP2	Tissue	Risk prediction signature: METT14, WTAP, HNRNPC, YTHDF1, and IGF2BP2.	[181]
Head and neck squamous cell carcinoma.	YTHDC2	Tissue	YTHDC2 correlates with prognosis and immune infiltration level (CD4+ T cell subpopulation). YTHDC2 has a possible tumour suppressor role.	[182]
Head and neck squamous cell carcinoma	IGF2BP2	Tissue	IGF2BP2 was identified as a hub m^6^A regulator, and its high expressions are correlated with poor prognosis.	[183]
Melanoma	ALKBH5, YTHDF1, KIAA1429	Tissue	Prognostic risk signature: ALKBH5, YTHDF1, and KIAA1429 divides patients into high- and low-risk OS groups.	[184]
Melanoma	YTHDF1, HNRNPA2B1	Tissue	Tumour stage and treatment response differ between patients with/without mutations in m^6^A regulatory genes.	[185]
Uveal melanoma	RBM15B, IGF2BP1, IGF2BP2, YTHDF3, YTHDF1	Tissue	m^6^A regulators with prognostic value: RBM15B, IGF2BP1, IGF2BP2, YTHDF3, and YTHDF1. RBM15B is an independent prognostic factor and correlates with clinicopathologic characteristics.	[186]
Osteosarcoma	KIAA1429, HNRNPA2B1 METTL3, YTHDF3, METTL14, FTO, YTHDF2	Tissue	Prognostic signatures.	[187]
Papillary thyroid carcinoma	HNRNPC, WTAP, RBM15, YTHDC2, YTHDC1, FTO, METTL14, METTL3, ALKBH5, KIAA1429, YTHDF1, ZC3H13	Tissue	Prognostic signature RBM15, KIAA1429, FTO.	[188]
Endocrine system tumours	IGF2BP1, METTL14, RBMX, HNRNPC, IGF2BP3, HNRNPA2B1 ICBLL1, RBM15B, KIAA1429, WTAP	Tissue	Prognostic signatures.	[189]
Adrenocortical carcinoma	RBM15, ZC3H3, YTDHF1, YTDHF2, ALBH5, KIAA1429, YTHDC1, HNRNPC, WTAP, METTL3, FTO	Tissue	Independent prognostic risk signature: HNRNPC, RBM15, METTL14, and FTO.	[190]
Clear cell renal cell carcinoma	METTL3, METTL14	Tissue	METTL3 and METTL14 are associated with prognosis and clinicopathological features.	[191]
Clear cell renal cell carcinoma	FTO, IGF2BP2, IGF2BP3, KIAA1429, YTHDC1, ZC3H13	Tissue	m^6^A-related risk signature for prognosis. m^6^A regulators’ expression correlates with histological grade and staging.	[192]
Clear cell renal cell carcinoma	ALKBH5, FTO	Tissue	ALKBH5 and FTO decreased gene expression correlates with poor OS.	[193]
Clear cell renal cell carcinoma	METTL14	Tissue	METTL14 probably methylates m6A in PTEN, leading to its expression change. METTL14 gene expression negatively correlates with the tumour stages and positively correlates with KIRC patients’ OS.	[194]
Bladder cancer	HNRNPA2B1 IGF2BP1, IGF2BP3, METTL3, YTHDF2, YTHDF1, FTO, ZC3H13, YTHDF3, YTHDC1, WTAP, METTL16, METTL14	Tissue	Identified risk factors that correlate with advanced clinical stages: RBM15, HNRNPA2B1, HNRNPC, IGF2BP2, YTHDF1, and YTHDF2.	[195]
Bladder cancer	METTL3, WTAP, FTO, YTHDC1	Tissue	Independent prognostic signature and predictor of clinicopathology.	[196]

**Table 4 cancers-15-01232-t004:** Diverse m^6^A regulators and the direction of their expression in different glioma grades with a possible role in diagnostics (these regulators do not have contradictory expressions in the studies presented in Table 3). Regulators in **bold** were detected in more than one study; relative increases and decreases in m^6^A regulator levels are indicated by up and down arrows, respectively.

m^6^A Regulators	GBM vs. NT	GBM vs. AST	GBM vs. LGG	High vs. Low Grade	Refs.
**FTO**	↓	↓	↓	↓	[100,105,107]
**METTL3**	↑	↓	↓		[100,101,107,205]
**RBM15**	↑				[100,105,107]
RBM15B	↑				[107]
**ZC3H13**	↓	↓	↓		[100,107]
KIAA1429 (VIRMA)			↓		[107]
**eIF3A**		↓		↓	[101,202]
eIF3B		↑	↑	↑	[202]
eIF3E			↑		[202]
eIF3I		↑	↑	↑	[202]
eIF3K		↑	↑	↑	[202]
eIF3L		↓		↓	[202]
eIF3M		↑	↑	↑	[202]
FMR1		↓			[101]
**HNRPC**	↑		↓		[100,107,213]
HNRNPA2B1	↑				[107]
IGF2BP2		↑			[101]
**IGF2BP3**	↑	↑		↑	[101,212]
RBMX		↓			[101]
**YTHDF1**	↑	↑			[100,101,107,213]
**YTHDF2**	↑				[100,107,213]
**YTHDF3**	↑	↓			[101,107,214]

Since this review is largely based on studies that do not strictly comply with WHO 2021 brain tumour classification [197], the glioma group names are defined based on those used in the included publications: GBM (IDH-wildtype and -mutant; note, GBM is now a designation of IDH-wildtype tumours only), astrocytoma (AST; IDH-mutant and -wildtype), low-grade glioma (LGG; different IDH status), and non-tumour brain tissue (NT). In addition, many cited studies do not include the specific glioma tumour grade.

## Data Availability

Not applicable.

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
