# Peer review of "Understanding the Epitranscriptome for Avant-Garde Brain Tumour Diagnostics"

_cancers, 2023, doi:10.3390/cancers15041232_

Round 1
Reviewer 1 Report
In this manuscript, the authors summarized the current understanding of the regulators and mechanisms of several major RNA modifications, with a focus on m6A, and reviewed recent findings on the roles of such modifications and their regulators in neurobiology and tumors as well as their diagnostic value. Overall, the review is comprehensive, well-organized, and well-written. Several grammar errors/typos can be seen in the manuscript, with a few examples below:
Line 100: “The m6A ‘writer’ complex (is comprised of…” should be “The m6A ‘writer’ complex is comprised of…”.
Line 105: “RNA, specifically binding to the Rm6ACH motif…” should be “RNA by specifically binding to the Rm6ACH motif…”.
Line 110: “and appears to be essential…” should be “and appear to be essential…”.
Line 243: “different to what was encoded by…” should be “different to what is encoded by…”
Line 295: “where OPC numbers are within normal limits” should be “where OPC numbers were within normal limits”.
Author Response
Several grammar errors/typos can be seen in the manuscript:
We have made several amendments to correct typographical/grammatical errors, including those mentioned in lines 100, 105, 110, 243 and 295. Please note, when viewing 'all mark up' in tracked changes the line numbers are different.
Reviewer 2 Report
The review submitted by Tűzesi et al. seeks to place and characterize the emerging field of epitranscriptome profiling in the context of neurological disease and, in particular, of glioma diagnostics. The proposed subject is interesting and may be of relevance for researchers in the field, but the manuscript needs to be improved in order to be suitable for publication in Cancers.
The main flaw of this review is that the references cited are not up to date. There are no cited references from 2022, with the only exception of reference [118], related to COVID-19. Given the rapid advances in RNA modification measurement technologies and the increasing number of contributions concerning epitranscriptomics in cancer, a one-year gap in citations is already too long. In addition, out of 168 references, about 88% are prior to 2021.
The part of the work devoted to technologies and methods for analyzing epitranscriptome (lines 428 to 438) is too concise, not detailed and not updated to current times. It is mandatory to include recent references and deepen this part well if the authors speak of ‘avant-garde diagnostics’ in the title. Moreover, given the ferment of bioinformatics work in this area, the sentence in line 435 ‘bioinformatic pipeline are in their infancy, and data analysis and interpretation is complex’, citing an old reference (Zaccara et al. 2019) does not seem appropriate.
At the end of the article (line 548) the authors speak for the first time of ‘resolution and detection sensitivity of RNA modifications’; if they want to mention these aspects perhaps they should add more in-depth considerations within the description of the different epitranscriptome profiling technologies and not just write a sentence in discussion.
Minor:
In Figure 1 caption, at line 81, substitute ’I to A‘ with the extended form, it should be ‘Adenosine-to-Inosine (A-to-I)’ as at this point of the manuscript it is still not defined.
Typos: ‘mainly’ line 132; ‘binding’ line 154
Remove comma after similarly at line 238: “similarly to DNA, ...” instead of “similarly, to DNA, ...”
Author Response
- The main flaw of this review is that the references cited are not up to date. There are no cited references from 2022…
We have updated the manuscript with multiple (30+) studies published in 2022. Many of these references are included in Table 2, Expression of epitranscriptomic regulators and their observed roles in diverse conditions.
Some recent studies specifically relate to glioma and we have discussed these findings in the text (lines 713-5, 772-6, 736-9, 1061-2, 1071-3)
- The part of the work devoted to technologies and methods for analyzing epitranscriptome (lines 428 to 438) is too concise, not detailed and not updated to current times. It is mandatory to include recent references and deepen this part well if the authors speak of ‘avant-garde diagnostics’ in the title…
We have included a new subchapter (4.1 RNA methylation detection methods, lines 1075-1486) that provides a more detailed overview of existing RNA methylation detection methods and contextualises the need for technological advances to adapt epitranscriptomic approaches for clinical brain tumour diagnostics.
- At the end of the article (line 548) the authors speak for the first time of ‘resolution and detection sensitivity of RNA modifications’; if they want to mention these aspects perhaps they should add more in-depth considerations within the description of the different epitranscriptome profiling technologies and not just write a sentence in discussion.
We appreciate this point and have revised the discussion to include a more in-depth description of the different epitranscriptomics profiling approaches (section 4.1). While direct assessments of RNA modifications in diagnostic laboratory settings are still technologically challenging, we sought to provide a thorough overview of the RNA modifications and epitranscriptomic regulators relevant to glioma biology and, and highlight their potential use as clinically relevant biomarkers.
Minor:
In Figure 1 caption, at line 81, substitute ’I to A‘ with the extended form, it should be ‘Adenosine-to-Inosine (A-to-I)’ as at this point of the manuscript it is still not defined.
The error in the Figure 1 caption at line 81 was corrected.
Typos: ‘mainly’ line 132; ‘binding’ line 154
The typos in lines 132 and 154 were corrected.
Remove comma after similarly at line 238: “similarly to DNA, ...” instead of “similarly, to DNA, ...”
The typo in line 238 was corrected.
Round 2
Reviewer 2 Report
The review submitted by Tűzesi et al. characterizes the emerging field of epitranscriptome profiling in the context of neurological disease and, in particular, of glioma diagnostics.
The proposed subject is interesting and of relevance for researchers in the field. I appreciated the modifications implemented by the Authors, especially the insertion of the paragraph 4.1, the insertion of recent references and the revision of the English which further improved the readability of the work.
The revised manuscript is suitable for publication in Cancers. I write below a few last suggestions.
Typos:
‘require’ line 513; ‘facilitate’ line 548
I would add some references for lines 516-519.
Author Response
Thank you for the additional comments. We have corrected the two typographical errors and have cited a further 3 references detailing NGS-based detection methods [128-130].